5

# **Unveiling the Link Between Extreme Precipitation Events and Flood Disasters in China: From a 3D Perspective**

Jie Wang<sup>1,2,3</sup>, Sixuan Li<sup>4</sup>, Xiaodan Guan<sup>1,4\*</sup>, Yongli He<sup>1,4</sup>, Chenyu Cao<sup>4,5</sup>, Lulu Lian<sup>1,4</sup>, Lihui Zhang<sup>1,4</sup>

<sup>1</sup>Collaborative Innovation Center for Western Ecological Safety, Lanzhou University, Lanzhou, China

Correspondence to: Jie Wang (wang\_jie@lzu.edu.cn)

Abstract. Extreme precipitation events and their triggered flood disasters have received increasing attention owing to their severe threats to human lives and socioeconomic development. However, there is still a lack of research on their evolutionary characteristics and driving factors from a three-dimensional (3D) event-based perspective. Here, we developed a 3D automatic recognition algorithm based on the connected component 3D algorithm. This method was applied to investigate the 3D characteristics of 632 flood-causing precipitation (FCP) events in China from 2000 to 2023. The associated flood disasters and their underlying driving factors were further analysed. The FCP events with larger accumulated magnitudes and affected areas are mainly distributed in the center of Southern China (SC) and Northern China (NC), mostly moving eastward with longer distances and lifespans. FCP-induced flood disasters are more severe in the SC and parts of the NC, while a relatively higher proportion of flood disaster losses are concentrated in the southeastern fringe of the Oinghai-Tibetan Plateau (TP) and southwestern China (SWC). In other words, flood disasters caused by FCP in China exhibit the changing characteristics of "high impact-low losses ratio" in SC and NC and "low impact-high losses ratio" in TP and SWC. Notably, despite the increase in 3D characteristics of FCP events over the past two decades, flood disasters have shown a significant reduction, except for the direct economic losses. Driving factor analysis indicates that the combination of precipitation and environmental factors have the greatest explanatory power for most flood disasters in China, while human activities have a prominent impact on the flood disasters in the center of SC and NC. These findings provide new insights into the characteristics of FCP events and their associated flood disasters from a 3D event-based perspective.

## 1 Introduction

Flooding induced by extreme rainfall is one of the most hazardous natural disasters, posing significant threats to human security and socioeconomic development worldwide (Jevrejeva et al., 2018; Rentschler et al., 2022). Over the past decade, under the context of global warming and an intensifying hydrological cycle, the frequency and magnitude of destructive floods have escalated due to the increased frequency of extreme precipitation (Blöschl et al., 2019; Wasko et al., 2021),

<sup>&</sup>lt;sup>2</sup>Key Laboratory of Urban Meteorology, China Meteorological Administration, Beijing, China

<sup>&</sup>lt;sup>3</sup>China Meteorological Administration Hydro-Meteorology Key Laboratory, Beijing, China

<sup>&</sup>lt;sup>4</sup>School of Atmospheric Sciences, Lanzhou University, Lanzhou, China

<sup>&</sup>lt;sup>5</sup>College of Earth and Environmental Sciences, Lanzhou University, Lanzhou, China

35

55

resulting in immense economic devastation and loss of human lives (Rentschler et al., 2022; Islam and Wang, 2024). According to the Emergency Events Database (EM-DAT) of the Centre for Research on the Epidemiology of Disasters (CRED), a total of 4,845 flood disasters occurred from 1980 to 2020, causing average annual economic losses exceeding USD 21 billion and approximately 5,900 deaths per year. Therefore, rainstorm-induced flood disasters have become a crucial issue restricting the sustainable and healthy development of global economy and society.

Heavy precipitation is a prominent factor leading to floods, and its quantity, intensity, and duration determine the flooding process in regions where rainfall plays the dominant role in flood occurrence (Mallakpour and Villarini, 2015; Do et al., 2020b). Extensive research has examined the changes of extreme precipitation at regional and global scales using a range of indices and detection methodologies (Westra et al., 2013; Asadieh and Krakauer, 2015; Ban et al., 2015; Wu et al., 2019; Chinita et al., 2021). These studies reported that the frequency and intensity of heavy precipitation have increased. Some research using rain gauge observations suggests that sub-daily heavy precipitation may have increased more than daily heavy precipitation (Chinita et al., 2021). The concerns about such superadiabatic increases have further been raised by modeling simulation, which projected further to increase between 2 % and 10 % by 2100 under a high-emissions scenario (Kharin et al., 2013). Notably, the heaviest and rarest precipitation events are projected to have the largest increases in frequency and intensity (Thackeray et al., 2022). However, despite the observed and projected increase in extreme precipitation, the flood magnitude and frequency exhibit mixed trend patterns. Several global scale flood trend detection studies have found that more stations exhibit significant decreasing trends in flood magnitude than increasing ones (Do et al., 2017; Hodgkins et al., 2017; Wasko et al., 2021). These inconsistencies highlight the nonlinear relationship between precipitation intensity and flood disasters, driven by multiple interacting factors such as land use, hydrological capacity, and human interventions (Sharma et al., 2018; Do et al., 2020a). Therefore, understanding how changes in extreme precipitation translate into flood impacts remains a critical research challenge.

Several previous research have attempted to quantify the relationship between heavy precipitation and flood damage using statistical analysis and hydrological model simulations (Kundzewicz et al., 2014; Wei et al., 2018; Davenport et al., 2021; Rashid et al., 2023; Teale and Winter, 2024). For instance, Kundzewicz et al. (2014) reviewed evidence showing that changes in precipitation contribute to increased flood damages. Similarly, Davenport et al. (2021) estimated that increasing heavy precipitation contributed approximately 36 % of cumulative flood damages from 1988 to 2017 in the U.S. Looking forward, Rashid et al. (2023) developed a probabilistic model based on the probability of property damage and heavy precipitation indicators, indicating an increase in high property damage in the future. However, an extreme precipitation event often propagates across both space and time during flooding periods (Wang et al., 2025). Most previous studies have focused on isolated extreme precipitation properties (e.g., magnitude and intensity) without considering the impact of comprehensive spatiotemporal evolution characteristics of precipitation systems (e.g., moving direction, distance, and lifespan), which can critically influence flood damages. This gap limits our ability to quantify the impact of heavy

precipitation characteristics on flood damages within individual precipitation events.

China has been the country most affected by hydrological and meteorological disasters, suffering from significant economic losses and numerous fatalities over the past several decades (Wei et al., 2018). Many studies examined the spatial-temporal characteristics of flood damages and evaluated the relationships between rainfall and flood damages, but few have investigated these characteristics from an event-based evaluation perspective (Li et al., 2012; Chen et al., 2021; Jia et al., 2022; Wang et al., 2022a). Thus, more systematic, and reasonable studies are urgently needed in this aspect. Here, we perform the first study to improve understanding of how changes in the evolutionary characteristics of heavy precipitation impact flood damages at an event perspective using a large precipitation sample (1,042) and the flood damage metrics including destroyed cropland area, direct economic loss, affected population, and death population during rainfall period from 2000-2023 across China. In doing so, we used connected component three-dimensional (3D) algorithm, similar to that of Wang et al. (2025), to explore the evaluation characteristics of flood-causing rainstorm events in China. The flood damages caused by each rainstorm were then analysed for individual precipitation events. Finally, the underlying driving factors of flood disasters were assessed from an event-based perspective. This study will provide a deeper understanding of historical flood risk trends and enable accurate prediction of fatalities and losses based on the precipitation characteristics.

## 2 Data

## 2.1 Satellite-based precipitation data

To analyse the spatiotemporal evolution patterns of rainstorm events over mainland China, we adopted the Integrated Multi-satellite Retrievals for Global Precipitation Measurement (GPM) (IMERG) precipitation product provided by the U.S. GPM project. The IMERG precipitation product integrates all active and passive microwave (PMW) precipitation estimates from the GPM constellation, fills temporal gaps with interpolated PMW data and geostationary infrared (IR)-derived data, and is calibrated using the monthly Global Precipitation Climatology Centre (GPCC) gauge analysis data (Huffman et al., 2020). The performance of IMERG has been extensively evaluated at numerous global locations, demonstrating its potential to capture the precipitation detection capability (Tang et al., 2020; Pradhan et al., 2022). Notably, the latest Version 07, released in 2023, introduces a wide range of improvements, including updated input algorithms, improved inter-calibration of PMW data, corrections of spatial offsets, and refined data processing procedures (see Huffman et al. (2023) for details). These improvements enhance precipitation detection capability, particularly for snowfall and extreme precipitation (Xiong et al., 2025). In this study, we used the IMERG half-hourly Final Run V07 product with high spatial (0.1°) and temporal (0.5 h) resolution covering the period 2000 to 2023. The "precipitationCal" variable, calibrated with gauge stations, was chosen. All IMERG product levels used in this study are accessible via the NASA website (https://gpm.nasa.gov/).

## 2.2 Flood disaster data

Historical flood damage data in 31 provincial administrative regions in China from 2000 to 2023 were obtained from the Meteorological Disaster Yearbook (MDY) in China. The data primarily include the flood records (i.e., flood location,




affected area, and date of the event) as well as corresponding direct economic losses, the number of people affected, the number of deaths, and the affected crop area. The MDY of China is currently the most comprehensive freely available source for accessing the flood records in China, and has been widely used in previous studies (Li et al., 2012; Wei et al., 2018; Shi et al., 2020; Wang et al., 2025). Unfortunately, this dataset has not been updated since 2020, thus, we supplemented flood disaster data from 2020 to 2023 using news reports and government sources by searching for the keywords 'flood' and 'inundation' online.

A total of 1,041 flood disaster records were obtained from MDY. The spatial-temporal distribution of these flood occurrence records across mainland China from 2000 to 2023 are depicted in Fig. 1. The accumulated flood occurrences frequency generally shows a decreasing trend from southeast to northwest (Fig. 1a), which is consistent with the distribution pattern of annual precipitation in China (Ma et al., 2015). The temporal variation of flood records in Fig. 1b shows that more flood disasters occurred during 2002-2010.

**Figure 1.** (a) Spatial distribution of accumulated flood records in China during 2020-2023. (b) Heat map of temporal variation of flood records for each province in China during 2020-2023. The characters in Fig. 1a denote the abbreviations of Chinese provincial names. The red lines indicate the division of four subregions: Northwestern China (NWC, purple font), Northern China (NC, blue font), Southern China (SC, red font), and Qinghai-Tibetan Plateau (TP, green font).

## 2.3 Flood disaster influencing factors

Meteorological, environmental, and anthropogenic factors are the main drivers affecting flood occurrence in any area. Among meteorological conditions, rainfall is typically the primary cause of flood disasters. In our study, the 3D precipitation features (including accumulated area, accumulated magnitude, lifespan, and moving distance, detailed descriptions in Section 3.3) were selected to determine their influence on the spatial distribution of flood disasters. The environmental factors are important components of the disaster-forming environment (Ke et al., 2025). We selected cropland, normalized difference vegetation index (NDVI), slope, and elevation differences to represent cropland area, vegetation coverage,

https://doi.org/10.5194/egusphere-2025-4728 Preprint. Discussion started: 10 November 2025

gradient of the land surface, and elevation variations, as the environmental factors in this study. Additionally, human activity is an important cause of altering the natural environment and influencing flood disasters (Guan et al., 2021). Based on the existing research (Wu et al., 2021; Hoang and Liou, 2024), the population and gross domestic product (GDP) are selected as important indicators to reflect human activity. All environmental and human activity factors were obtained from the Resource and Environment Data Cloud Platform (http://www.resdc.cn), except for cropland, which was sourced from https://doi.org/10.5281/zenodo.7936885 (Tu et al., 2024). The original data were interpolated with the same spatial resolution of the precipitation with 8 km. In addition, it is worth noting that the population and GDP were summed within grids cells. Meanwhile, due to the lack of annual data, population and GDP data for 2000-2010 and 2010-2020 were represented by the 2005 and 2020 datasets, respectively. A summary of all variables and their data sources is provided in Table 1.

## 130 **3 Methodology**




## 3.1 Definition of precipitation threshold values

Following the definition of short-duration heavy rainfall events by the China Meteorological Administration (CMA), an hourly rainfall amount exceeding 16 mm is commonly defined as a heavy rainfall event, which has been validated by (Zhang and Zhai, 2011) to be a reasonable criterion to study short-duration heavy rainfall events in China. However, the significant spatial-temporal variations may occur in extreme events as a result of varying geographical and meteorological conditions and consequently, the threshold of extreme precipitation events varies across regions. For instance, hourly extreme precipitation exceeding 16 mm/h is relatively more frequent over the southern China but it would be rather rare in the northwestern China (Fig. 2b). Thus, a fixed threshold cannot be used to extract relatively independent extreme precipitation events over China. In previous studies, most studies defined the extreme precipitation events using the 90 % or 95 % percentile of the ordered precipitation sequence during the study period because of its simplicity (Gu et al., 2022). In this study, the 95 % percentile of precipitation sequence at each grid was defined as the thresholds for extreme events (Fig. 2c). Hourly precipitation amounts exceeding the extreme precipitation threshold were counted an extreme precipitation event. Figures 2d and 2e show the spatial distribution of the amount and accumulated frequency of extreme precipitation, respectively. Overall, extreme precipitation is observed across China, although more frequent in the southeastern regions than in northwestern areas. Subsequently, these threshold values were subsequently used to better extract the characteristics of contiguous extreme precipitation events.

**Figure 2.** Spatial distributions of extreme precipitation amount (a and d) and frequency (b and e) calculated using the single threshold (>16 mm) in first row and 95 % percentile (Fig. 2c) in second row, respectively.

## 150 3.2 Identification of flood-causing precipitation event



Historical flood damage sample data records flood events at country level in each province and provides the location and date of each flood event from 2000 to 2023. However, a flood-causing precipitation event mostly occurs synchronously in neighboring regions or consecutively in adjacent days, leading to multiple flood records during a single precipitation process. In our study, we propose a precipitation event identification algorithm to identify relatively independent extreme precipitation event during the study period of each event. The basic steps of the precipitation event identification algorithm are described below (Fig. 3):

(1) Determining the start and end times of FCP event: For each event  $(E_i)$ , the approximate occurrence time  $(t_i)$  is obtained from the flood disasters dataset. Based on the previous research findings, the lifetime of an FCP event is generally not greater than 3 days, and it is rare for such events to last more than 7 days (Wang et al., 2022a). Thus, in order to detect all potentially associated persistent FCP events, we extract the FCP identification period for the 5 days before and after  $t_i$ . Subsequently, we used the vector shapefile of the flood-affected areas  $(l_i)$  as a spatial mask to isolate IMERG precipitation



within the affected area during identification period. Based on the precipitation time-series, we can accurately determine the start and end time of FCP event.

165 **Figure 3.** Flowchart of the precipitation event identification algorithm.

(2) Determining the FCP event in flood-affected area: The accumulated precipitation during start and end time of the FCP event  $(E_i)$  is first marked when its accumulated amount (P) exceeds the threshold value. The threshold value is the mean of 95 % percentile of hourly precipitation amounts within flood-affected area. The 95 % percentile value is computed from all rainy hours (P > 0.1 mm) in the rainy season (from May to October) of 2000-2023. In our study, grid locations with less than threshold values of accumulated amount of FCP event  $(E_i)$  are excluded from our analysis. Subsequently, we stored the hourly precipitation during start and end time of FCP event  $(E_i)$  in a 3D array of voxel array (latitude×longitude×time) in "0/1" binary format. Here, if there is a rainy hour at the pixel location marked in the last step, it will be labeled "1"; otherwise, it will be labeled "0". This detection method is based on the connected component 3D (CC3D) algorithm (Silversmith, 2021). The CC3D algorithm code is the freely available Python package connected-components-3d (available at https://pypi.org/project/connected-components-3d/). The "0/1" binary array is imported into the CC3D algorithm to identify all possibly connected voxels by setting connectivity to 26. The 26-connectivity searching allows that a contiguous precipitation event occurring at a grid on the current hour can move to the adjacent grids in the following hour, and ensures that different contiguous precipitation events  $E_i$  do not overlapped or touched in the space, that is, different events are neither spatially adjacent nor temporally contiguous. As a result,  $E_i$  contiguous precipitation events are retained during start and end time of FCP event  $(E_i)$ . If the flood-affected area is covered by the contiguous precipitation event  $E_i$ ,  $E_i$  is classified as a flood-causing precipitation event.

190

(3) Determining the independence of FCP event: To ensure that the FCP event ( $E_{ij}$ ) is a single event, meaning  $E_{ij}$  is neither spatially adjacent nor temporally contiguous, we utilized the CC3D algorithm to forward-detect the next 10 flood records (i.e.,  $E_{ij+1}, \dots, E_{ij+10}$ ) for each  $E_{ij}$ . If any part of  $E_{ij}$  not spatial overlaps with any part of events (i.e.,  $E_{ij+1}, \dots, E_{ij+10}$ ),  $E_{ij}$  is considered a single event. Meanwhile, if the overlap ratio exceeded 50 % between event  $E_{ij}$  and other events (i.e.,  $E_{ij+1}, \dots, E_{ij+10}$ ), but the time interval between the events is more than 2 days, and they are from different precipitation types (e.g., TC and non-TC), the  $E_{ij}$  is considered as an independent event.

In this study, based on the above FCP identification processes, 1,041 flood disaster records were classified into 632 independent precipitation events, including 368 non-TC events and 246 TC events.

## 3.3 Calculating the 3D structure properties of contiguous FCP event

For each contiguous FCP event identified by the previous step, the 3D structural properties can be characterized by the following multidimensional metrics, and shown in Figs. 4a-b, consistent with previous studies (Wang et al., 2022a; Wang et al., 2025):

195 **Figure 4.** (a) Three-dimensional (3D) evaluation of a spatiotemporally contiguous FCP event occurred on 21 July 2021 in Henan of China. (b) The evaluation features of FCP event. The shading indicates the FCP accumulated area. The red dot indicates the centroid of contiguous FCP event. The red arrow represents the movement of FCP event, the moving direction is from the tail to the head of the arrow, and the length of arrow indicates moving distance. (c) The human activity factors (including population and GDP) in the heavy rainfall coverage areas of FCP event occurred in 21 July 2021 in Henan of China. (d) The environment factors of Earth's surface, including cropland, NDVI, slope, and elevation difference, in the heavy rainfall coverage areas of FCP event occurred on 21 July 2021 in Henan of China.

© Author(s) 2025. CC BY 4.0 License.

- (1) Accumulated magnitude: The sum of the precipitation amount over all grid cells across all hours of the FCP event.
- (2) Accumulated affected area: The projected area on the land surface affected by the FCP event.
- (3) Centroid: Represents the position of precipitation in the 3D space (latitude×longitude×time), calculated as the weighted center of FCP event within 3D space.
  - (4) Lifespan: The time interval from the centroid at the start to the centroid at the end of the FCP event.
  - (5) Moving direction: The azimuth angle pointing from the centroid of the begin time to the centroid of the end time.
  - (6) Moving distance: The maximum distance across all hourly centroids during FCP event period.

# 3.4 Geographical detector model

The geographical detector model is a set of statistical methods to explore spatial heterogeneity in explanatory variables, reveal interactions between variables, and quantify contribution rate of the driving factors (Wang and Hu, 2012). The formulas for ecological detection, risk detection, interaction detection and factor detection are given in Wang et al. (2010). The core idea of factor detection is to determine whether the independent variable can explain the dependent variables and to investigate whether the interaction of two independent variables enhances, diminishes, or has no effect on the dependent variable. The results of interaction detector between two independent variables are classified into 5 types in Table 2. By comparing the q value of each factor and the q value of the two factors superposition, the geographical detector can determine whether there is an interaction between two factors, and whether their interaction is linear or nonlinear. Generally, q ∈ [0, 1], q value is close to 1, meaning the independent variables has a strong influence on independent variables, while a q value is close to 0, indicating that the dependent variable is spatially randomly distributed. The detailed description of geographical detector model can be found in Wang and Hu (2012). The formula for the q value is depicted as follows:

$$q = 1 - \frac{1}{N\sigma^2} \sum_{i=1}^n N_i \sigma_i^2 \tag{1}$$

where,  $i=1,\cdots,n$  represents the stratification for variable factor; N and  $N_i$  denote the total number of the units in the whole region and in stratum i, respectively;  $\sigma_h^2$  and  $\sigma^2$  are the variances of Y in  $N_i$  and N, respectively.

Referring to Liu et al. (2018), 10 variables related to precipitation, the land surface environment, and human activities were selected to determine the main factors influencing the flood disaster spatial distribution. The precipitation variables can be characterized by the 3D structure properties of contiguous FCP event, including accumulated area, accumulated magnitude, lifespan, and moving distance. It is noted that the environment of the land surface (i.e., cropland, NDVI, slope of terrain, and elevation differences of terrain) and human activities (i.e., population and GDP) variables are represented by average values within the FCP event coverage areas, as illustrated in Figs. 4c and 4d.

## 230 3.5 The scatter degree method

The scatter degree method (SDM) is an objective evaluation method that reflects the overall differences among evaluated objects. The basic principle of SDM is to select weight coefficient for indicators so that the difference among evaluated




objects is maximized. Therefore, deriving weight coefficients for each indicator based on their intrinsic importance is critical. In our study, the SDM is employed to determine the weight of each indicator and calculate the comprehensive evaluation index for flood disaster and their driving factors, as the following steps:

Assume that the comprehensive evaluation function for  $\nu$  can be expressed as:

$$y = \omega_1 o_1 + \omega_2 o_2 + \dots + \omega_m o_m = \mathbf{w}^{\mathsf{T}} \mathbf{0}$$
 (2)

where,  $\mathbf{w} = (\omega_1, \omega_2, \dots, \omega_m)^T$  denotes the weight coefficient vector,  $\mathbf{0} = (o_1, o_2, \dots, o_m)^T$  represents the standardized evaluated object vector.

According to the SDM method, let  $\mathbf{H} = \mathbf{0}^T \mathbf{0}$ , which is a real symmetric matrix. By calculating the eigenvalues of  $\mathbf{H}$ , we can 240 obtain the standard feature vector of maximum eigenvalue. Finally, the weight  $\omega_i$  for each sub-risk indicator is obtained from this eigenvector. Subsequently, put  $\omega_i$  into Eq. (2), the comprehensive evaluation index of flood disaster and their driving factors is computed.

## 4 Results

#### 245 4.1 Spatial distribution of contiguous FCP events in China

Significant contrasts exist between TC and non-TC rainstorm induced flood characteristics (Lu et al., 2020). In our study, we first categorized 632 FCP events into 246 TC and 368 non-TC precipitation events, and provided their 3D structural properties across China in Fig. 5. The centroids of an FCP events represent the geographical location, as shown in Figs. 5a and 5b. The size and color of the centroid denote the accumulated area and magnitude, respectively. TC-induced rainstorm events mostly occurred in the coastal areas of China, with the majority concentrated in coastal regions of Southern China (SC) rather than Northern China (NC) (Fig. 5a). These events have an average affected area of  $1.33 \times 10^2 \, km^2$ , and accumulated magnitude of  $5.65 \times 10^4$  mm. Notably, TCs that moved farther inland exhibited larger accumulated areas and magnitudes, especially in the northeastern part of NC.

Figure 5c shows the spatial distribution of the migration of TC-induced FCP events. The colors represent the lifespan of the 255 FCP events, the length of the arrows represents migration distances of FCP events, and arrow is the direction of the migration of FCP event. On average, TC-induced FCP event persists for 23.37 hours and travels 417.88 km. TC-induced FCP events are more inclined to move to the westward, which is consistent with an earlier study (Zhang et al., 2013). The westward shift of the western Pacific subtropical high (WPSH), westward steering flow, and easterly vertical wind shear are the main drivers of the formation of the TC moving westward. Meanwhile, some TCs prefer to shift northward or northeastward due to the eastward shift of WPSH, westerly vertical wind shear, weak mountainous blocking and westerly steering flow surrounding the TCs (Wang et al., 2022a).

Figure 5b shows the centroids of non-TC-induced FCP events. It can be observed that non-TC-induced FCP events are mainly clustered in Southern China and Northern China. The average accumulated magnitude and projected area are 10.47



 $\times$  10<sup>4</sup> mm and 2.77  $\times$  10<sup>3</sup> km<sup>2</sup>, respectively. The non-TC events with larger accumulated magnitude and projected areas are mainly distributed in the center of SC and NC. In contrast, the magnitude and affected area of non-TC events is larger than those of TC events. The migration characteristics of non-TC events are illustrated in Fig. 5d. It is seen that the non-TC events are more inclined to move eastward, which is mainly attributed to the mid-latitude climatological westerlies and southwesterly monsoon circulation. These eastward moving FCPs have an average traveling distance of 621.69 km and lifespan of 24.84 hours, which is slightly longer than TC-induced events. This is consistent with research by Wang et al. (2022a), who suggested that FCP events with more persistent lifespan tend to travel longer and exhibit larger accumulated magnitudes and projected areas.

**Figure 5.** (a) and (b) show the spatial distribution of the overall centroids of TC and non-TC flood-causing precipitation events in China during 2000-2023, respectively. The circle color and size indicate their accumulated magnitude and area, respectively. (c) and (d) depict the spatial movement of the TC and non-TC flood-causing precipitation events in China, respectively. The colors represent the lifespan. The arrow indicates the movement direction. The rose diagram illustrates the directional distribution of precipitation events in China, with color indicating the ratio of current direction to total events. The shading indicates the terrain height.



# 4.2 Temporal changes of contiguous FCP events in China

Figure 6 illustrates the long-term trends of multidimensional metrics of contiguous FCP events in China during 2000-2023. All metrics of contiguous FCP events show positive trends, which is consistent with previous studies on the Li and Zhao (2022). This is likely due to the increasing water vapor-holding capacity in the atmosphere under global and regional warming (Allan et al., 2022). The accumulated area and magnitude of FCP events show slight upward trends, with an annual variation of  $0.86 \times 10^2 \, km^2$ /year (P 

**Figure 6.** Time series of annual mean values of flood-causing precipitation (FCP) event metrics in China during 2000-2023: (a) accumulated area, (b) accumulated magnitude, (c) lifespan, and (d) moving distance. The blue solid line and straight line indicate the time evolution and trend of mean values of total FCP during 2000-2023, respectively. The blue shading indicate the  $\pm$  standard deviation across the evolution of total FCP events. The black dots and dashed line represent the annual mean


values and trend of non-TC FCP events during 2000-2023. The red triangles and dashed line represent the annually mean values and trend of TC FCP events during 2000-2023.

# 4.3 Spatial distribution of flood disaster in China

Figure 7 illustrates the spatial distribution of the flood affected population, death, direct economic loss, and affected cropland areas in China during 2020-2023. The geographical location of each flood event is represented by the centroid of the corresponding precipitation events. It can be easily observed from Fig. 7a that FCP events occurring in the SC and parts of the NC are mostly associated with larger affected population, whereas the flood-affected population in NWC and Qinghai-Tibetan Plateau (TP) is relatively low. Different spatial patterns can be found for the number of deaths caused by FCP events (Fig. 7b) across China when compared to that of flood-affected population (Fig. 7a). Higher number of deaths are observed in the SC and NC, but the number of deaths is fewer in center of SC. Besides, the number of deaths is relatively large in the NWC and the surrounding regions of TP. The direct economic losses caused by FCP events are more severe in the southeastern and western parts of SC, as well as the central and northeastern regions of NC (Fig. 7c). For cropland, the affected cropland areas are mainly distributed in southeastern regions of SC and central and northeastern regions of NC (Fig. 7d). Besides, among these FCP events, TC-induced flood result in more severe disasters than non-TC-induced events.


Figure 7. Spatial distribution of (a) flood affected population, (b) death, (c) direct economic losses, and (d) affected cropland areas in China during 2000-2023. Bar chart illustrates the mean values of flood disaster of TC and non-TC FCP events. The shading indicates the terrain height.

Figure 8. Spatial distribution of (a) the ratio of flood affected population to the total population within the coverage areas of FCP event, (b) the ratio of death to the total population within the coverage areas of FCP event, (c) the ratio of direct economic losses to the total GDP within the coverage areas of FCP event, and (d) the ratio of affected cropland area to the total cropland area within the coverage areas of FCP event in China. Bar chart illustrates the mean values of the ratio of flood disaster of TC and non-TC FCP events. The shading indicates the terrain height.

To further investigate the flood-affected rate within the coverage areas of FCP events, we calculated the ratios of flood affected population, deaths, direct economic loss, and affected cropland areas to the total population, GDP, and total cropland within the coverage areas of FCP events, respectively (Fig. 8). As shown in Fig. 8a, high ratios of flood affected population to the total population within the coverage areas of FCP events is predominantly concentrated in SC, similar to the affected population distribution in Fig. 7a. However, high death rates are primarily distributed in the southeastern fringe of the TP and southwestern China (Fig. 8b). This phenomenon can be attributed to the high population density and robust economic development in eastern and southern China, and enhanced flood mitigation capacity to mitigate floods impacts by



construction of levees and flood-mitigated infrastructure (Fang et al., 2018). In contrast, mountainous areas in the southeastern fringe of the TP and southwestern China lack such infrastructure. FCP events in these regions often trigger sudden-onset flash floods accompanied by landslides and debris flows, severely threatening human lives (Chen et al., 2021). For the direct economic loss, the ratio to the total GDP is low in southeastern SC but relatively high in the western SC and northeastern NC. This suggests that FCP events caused significant economic losses in the western SC, rather than eastern SC with relatively high GDP. Regarding affected cropland, the smaller absolute affected cropland areas but larger ratio of affected cropland to the total cropland within the coverage areas of FCP event can be found in the coastal areas of SC. This is because this region frequently experiences TC rainfall processes, which often coincide with strong winds and storm surges, posing severe threats to the cropland. Overall, flood disasters caused by FCP in China exhibit the changing characteristics of "high impact-low losses ratio" in SC and NC and "low impact-high losses ratio" in TP and SWC.

**Figure 9.** The time series boxplot (left y-axis) of (a) flood affected population, (b) death, (c) direct economic losses, and (d) affected cropland areas in China from 2000 to 2023. The time series of ratio (right y-axis) of (a) flood affected population, (b) death, (c) direct economic losses, and (d) affected cropland areas to total population, GDP, cropland within the coverage areas of FCP events. The black dashed line represents annual mean ratio. The shaded gray area represents the 95 % confidence interval on the basis of the quadratic fitted line (black line). Annual mean values and trends of flood (red and blue show the non-TC and TC FCP events, respectively) affected population, death, direct economic loss, and affected cropland areas over China from 2000 to 2023 shown in four subregions.

365

370

375

# 4.4 Temporal changes of flood disaster in China

345 Figure 9 shows the time series of the flood disasters in China during 2000-2023. As shown in Fig. 9a, the flood-affected population and the ratio of affected population to total population within the coverage areas of FCP events show a clear decreasing trend. The TC-induced affected population declines by  $14.95 \times 10^4$  persons/year, which is larger than the non-TC event with  $12.09 \times 10^4$  person/year (significant at the 95 % confidence level). The number of deaths also shows a decreasing trend during the period, except for extremely severe floods in 2016 and 2019. Meanwhile, we note that the TCinduced deaths decreased significantly by about 1.27 persons/year (P 



**Figure 10**. Heatmap showing the Pearson's correlation coefficient (CC) between flood disasters (flood affected population, death, direct economic losses, and affected cropland areas) and flood driving factors (3D precipitation features (accumulated area, accumulated magnitude, lifespan, and moving distance, blue font), environment factors (cropland area, NDVI, slope, and difference elevation, green font), and human activities (population and GDP, red font) factors). Significance correlations are marked by asterisk, \* *P*



**Figure 11**. Interaction detection results of Geodetector analysis of driving factors: (a) for affected population, (b) for death, (c) for direct economic losses, and (d) for affected cropland areas. \* indicates two-factor nonlinear enhancement; \*\* means two-factor bilinear enhancement.

To further reveal the spatial characteristics of interactions effect among driving factors on flood disaster, we calculated the comprehensive evaluation index of three driving factors based on the SDM method, and mapped the spatial distribution of dominant factors of flood disasters in Fig. 12. We found that precipitation and environmental factors have the largest spatial distribution among all potential driving factors, suggesting that flood disasters are influenced by combining precipitation and environment factors in China. Meanwhile, the flood disasters events significantly influenced by the human activities are predominantly distributed in the center of SC and NC, which are densely populated regions in China, e.g., the middle-lower Yangtze River plain and the North China plain. This indicates that flood caused more severe disasters in densely populated areas, e.g., urban areas, which is consistent with our above findings in Fig. 10.

**Figure 12**. Ternary diagram shows the interannual trend of flood disasters is dominated by precipitation, environment, and human activities factors. The shading indicates the terrain height.

## 410 5 Discussion



# 5.1 Analysis of flood-causing precipitation event features from a 3D perspective

Rainstorm is one of the most frequent natural hazards by triggering flooding and severe hazards in the world, seriously impacting populations, property, and socioeconomic development, thus, have garnered significant attention of the international society (Kharin et al., 2013; Ban et al., 2015; Wasko et al., 2021). In previous studies, the behaviours of rainstorms, such as frequency, intensity, and duration, have been extensively examined using maximum rainfall indices in specific regions (Fu et al., 2013; Deng et al., 2018). For example, Fu et al. (2013) constructed extreme rainfall indices and analysed spatial and temporal changes of extreme rainfall in China. However, these approaches only identified the continuous rainstorm spatial features at two-dimensional (latitude×longitude) and examined temporal changes based on the time series of extreme rainfall index, while ignoring the synchronous spatiotemporal connections and independence of rainfall events. This oversight could lead to an irrational analysis of rainfall features over study area. To overcome this limitation, recently, Silversmith (2021) proposed an integrated spatial contiguity algorithm to characterize precipitation events at the spatial scale. Wang et al. (2022a) investigated the 3D behaviours of spatiotemporal contiguous extreme








precipitation events in China, meanwhile, Wang et al. (2025) analysed the 3D evolutionary features of spatiotemporally contiguous flood-causing precipitation events using CC3D algorithm. Thus, view from a higher-dimensional perspective, the identified 3D precipitation features not only retain the features recognized through a lower dimensional perspective but also provide a more visual and stereo demonstration of the spatial moving process of precipitation events (Feng et al., 2024). In this study, we attempted to extend the CC3D algorithm to investigate the 3D characteristics of precipitation events and conducted the first comprehensive assessment of flood disasters from the event-based perspective. To this end, we developed an automatic precipitation event recognition algorithm with the CC3D method as its core, as shown in Fig. 3. This method in our study provides a possibility to accurately determine the spatiotemporal contiguous flood-causing precipitation events and reveal their "true" 3D structures and evolutionary patterns (e.g., moving direction, distance, and lifespan) from 2000 to 2023 in China. Additionally, the flood disasters caused by FCP events were extracted within the FCP event coverage areas, as illustrated in Fig. 4c, which could provide more reliable estimates of the influence of FCP events. Therefore, based on the above characteristics of events, our findings comprehensively reveal the characteristics of the precipitation and flood disasters from an event-based perspective. The results show that the FCP events were classified into 368 non-Tropical Cyclone (non-TC) and 246 TC events. TC-induced events, predominantly occurring in the coastal regions of Southern China (SC), have relatively small accumulated areas and magnitudes and short lifespan and moving distance, mostly inclined to move westward. In contrast, non-TC events are concentrated in SC and Northern China (NC), exhibiting eastward migration and characterized by larger accumulated magnitudes and affected areas and have a longer lifespan and moving distance than TC events. Thus, our research provided a new perspective for revealing the evolutionary features of spatiotemporally continuous flood-causing precipitation events from a 3D event-based perspective.

## 5.2 Potential driving factors of flood disasters in China

Precipitation, as a decisive factor of flood events, has attracted extensive attention from academia and policymakers. Many scholars have explored the spatiotemporal variability of extreme precipitation and its potential drivers in China. They suggested that the frequency of extreme precipitation events showed increasing trends and the relative contribution of extreme precipitation to the total precipitation also increased in most parts of China (Wu et al., 2019; Gu et al., 2022; Wang et al., 2022a; Fu et al., 2023). These patterns are closely linked to the increase of the moisture-holding capacity of the atmosphere with higher temperature at approximately 7%/°C (Trenberth et al., 2003). In comparison with these previous studies, our study introduces a new evaluation method, and comprehensively assesses the multi-dimensional characteristics of spatiotemporal FCP events. The results indicate that the accumulated area, accumulated magnitude, lifespan, and moving distance of FCP events are mostly increased during 2000-2023 (Fig. 6). This result is generally consistent with the trend of precipitation variation in China, and suggests that extreme precipitation events are projected to become more intense.

In contrast to the increasing trends and intensifying strength of extreme rainfall, flood disaster shows very mixed trends in China due to differences in analytical areas, data, and methods. For instance, Xie et al. (2018) pointed out that the combination of more frequent and stronger extreme rainfall and the rapid expansion of urban agglomeration is more likely to







increase the flood risk in Nanjing metropolitan. Similar results were also found along the coast of China (Feng et al., 2023), the extreme precipitation events led to a corresponding increase in the risk of flooding. However, in our study, the flood disasters are inconsistent with extreme rainfall, and represents a significant reduction in the past twenty years in China, except for the direct economic losses (Fig. 9), which has been confirmed by Wei et al. (2018). From the perspective of spatial distribution, FCP events with larger accumulated magnitude and projected areas are mainly distributed in the center of SC and NC, and cause more serious disaster losses, but the flood loss proportions are largely reduced. This discrepancy might be caused by larger-scale hydraulic projects, such as dams and reservoirs. By the end of 2018, there are 98,822 reservoirs exist in China, with a total storage capacity of  $8.95 \times 10^{12} \, m^3$  (Data are derived from China Water Statistical Yearbook 2018), which may significantly modulate the flood risks. Tang et al. (2023) studied the impact of dams on the flood in the Yangtze River, and found that dams and reservoirs mitigated the extreme flood by contributing to 94 % of the water level changes. In contrast, mountainous areas in the southeastern fringe of the TP and southwestern China, where the accumulated magnitude and projected areas of FCP events are relatively small, flood disaster proportions are large, especially the death. That means flash floods triggered by extreme rainfall events, representing a severe hazard in low-GDP regions. The robustness of the finding was found in Chen et al. (2025). In summary, flood disasters in China have been categorized into high-impact and low-loss in plain areas of SC, and low-impact and high-loss situation in the mountainous areas of west.

However, flooding is a complex physical process involving interactions among hydrology, meteorology, and land surface features can be triggered by multiple mechanisms. Determining the factors that influence flood disasters from a multiperspective is helpful for identifying the coupling mechanism between factors and flood disasters. We found that the human activities (e.g., population and GDP) are the second most influential factor after extreme precipitation, which corresponds to previous literature reporting a positive relation between flood disasters and population, meaning that large populations in flood-affected areas are expected to have large flood risk exposure (Rogers et al., 2025). Our further examinations demonstrate the interaction between precipitation and environment factors displays significant nonlinear enhancement and bivariate enhancement effects, suggesting that precipitation situation and the topographical features of rainstorm area will have a major impact on the evolution of the flood-waterlogging process (Fig. 11). Notably, the flood processes in the center of SC and NC are more susceptible to human activities (Fig. 12), posing significant threats to lives and property.

## 5.3 Uncertainty and limitations

Although our study provided estimation of the multi-dimensional characteristics of contiguous FCP events and first investigated their associated flood disasters and driving factors from an event-based perspective, there are several uncertainties and limitations that require further attention. It is essential to note that the 3D extreme rainfall features extracted with the different threshold values may have a different accumulated magnitude, projected area, lifespan, and moving distance. For instance, the accumulated areas of extreme rainfall extracted with the 95 % percentile of rainfall as a threshold may be smaller than that extracted with the 90 % percentile of rainfall. How to accurately determine the







appropriate extreme rainfall thresholds for multiple types of rainfall events needs to be further explored in further research. Meanwhile, the frequency of flood events cannot accurately represent based on the current data resources in this study. Therefore, due to data limitations, only precipitation features are considered in this study, and the temporal variations of frequency of flood were not analysed. Furthermore, flooding is a complex physical process that is typically influenced by a combination of natural and human factors. As this paper only conducted a preliminary assessment of the relationships between flood disasters and driving factors, e.g., precipitation features, environment factors (cropland area, NDVI, slope, and difference elevation), and human activities factors (population and GDP), while there may be other factors that probably exert important impact on flood disasters, such as hydraulic engineering projects (e.g., dams and reservoirs in mountain areas and flood-control measures in urban, as highlighted by Zhao et al. (2022) and Qi et al. (2022), which are a very effective measure for mitigating the flood risk. Thus, it is necessary to integrate human activities and natural factors in further research to explore the mechanism of flood disasters. Overall, the 3D method can effectively capture the evolutionary patterns of contiguous FCP events. Strengthening research on extracting accurate precipitation characteristics and using more reasonable flood driving factors is expected to enhance our understanding of relationship between extreme precipitation and flood disasters in China.

## 6. Conclusions

Based on 1,041 flood-causing precipitation events during the warm season (from May to September) from 2000 to 2023 in China, this study developed an automatic precipitation event recognition algorithm using connected component 3D (CC3D) method. By employing this approach, we provided a detailed analysis of flood-causing precipitation (FCP) events and their triggered flood disasters from a 3D event-based perspective, and deeply investigated the complex relationship between flood disasters and precipitation, environment, and human activity factors. The research findings are summarized as follows.

A total 1,041 flood disaster records were classified into 632 separate precipitation events, including 368 non-TC events and 246 TC events. It is found that TC-induced rainstorm events mostly occurred in the coastal areas of China, with the majority concentrated in coastal regions of Southern China (SC). These events are more inclined to move westward, and have a relatively small accumulated areas and magnitudes and short lifespans and moving distances. In contrast, the non-TC-induced FCP events are mainly clustered in SC and Northern China (NC). They trend to move eastward, with relatively larger accumulated magnitudes and affected areas, and have a longer lifespan and moving distance than TC events. Our further examinations demonstrate that these FCP 3D features in China have significantly increased during 2000-2023 (*P* 




affected ratio within the coverage areas of FCP events reveals that the spatial impact of floods on population and mortality exhibit significant spatial heterogeneity, in other words, the high flood affected population proportion is predominantly concentrated in SC, while high death rates are primarily distributed in the southeastern fringe of the TP and southwestern China. In terms of the ratio of direct economic losses, the small ratio events are mainly distributed in the southeastern SC but relatively high events are in the western SC and northeastern NC. Regarding ratio of affected cropland, larger event can be found in the coastal areas of SC. Furthermore, our findings indicate that flood disasters represent a significant reduction in the past twenty years in China, except for the direct economic losses.

The results obtained by tracking the relationship between flood disaster and their driving factors reveal that precipitation features, especially accumulated magnitudes and lifespans of FCP events, are the most important driving factors, although precipitation factors and flood disasters exhibit distinct changing trends during 2000-2023. Meanwhile, the flood disasters events are significantly influenced by the human activities, but environment factors have the lowest explanatory power. However, the interactive influence of any two factors is greater than that of the individual factors. Specifically, the interaction between precipitation factors and environment factors displays significant nonlinear enhancement and bivariate enhancement effects. This means that although environmental factors have weak explanatory power on their own, the synergistic effects could exert the stronger influence due to their nonlinear characteristics at different spatial and temporal scales. It is worth noting that most flood disasters are mainly influenced by the combination of precipitation and environment factors in China, while flood disasters events significantly influenced by the human activities are predominantly distributed in the center of SC and NC.

Code availability. The source code for our figures is available upon request from the corresponding author.

**Data availability.** All observational and reanalysis data used in this work are publicly available. The historical flood disaster Yearbook of Meteorological Disasters in China are available to download https://navi.cnki.net/knavi/yearbooks/YQXZH/detail?uniplatform=NZKPT. IMERG-F products are accessible via NASA website (https://gpm.nasa.gov/). The NDVI, DEM, population and GDP dataset are derived from Resource and Environment Data Cloud Platform (http://www.resdc.cn). The cropland datasets are obtained from https://doi.org/10.5281/zenodo.7936885. 545

Author contributions. J.W.: Writing-original draft, Visualization, Methodology, Formal analysis, Data curation, Conceptualization, Funding acquisition. S.L.: Data curation, Software; X.G.: Funding acquisition, Writing-review, Supervision, Conceptualization. Y.H.: Supervision, Writing-review & editing. C.C.: Software. L.L.: Writing-review & editing. L.Z.: Writing-review & editing.

Competing interests. The contact author has declared that none of the authors has any competing interests.

**Disclaimer.** Publisher's note: Copernicus Publications remains neutral with regard to jurisdictional claims made in the text, published maps, institutional affiliations, or any other geographical representation in this paper. While Copernicus Publications makes every effort to include appropriate place names, the final responsibility lies with the authors.

Acknowledgments. This material is based upon work supported in part by the National Natural Science Foundation of China (42205193), the Open Foundation of Key Laboratory of Urban Meteorology (LUM-2025-14), the Funding for High-Level Talents in Lanzhou City (563225108), the Open Foundation of China Meteorological Administration Hydro-Meteorology Key Laboratory (23SWQXM001), the Ministry of Education's Humanities and Social Sciences Youth Fund (24YJCGJW008), and the Fundamental Research Funds for the Central Universities (lzujbky-2025-jdzx01).

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
