# Peer review of "Unveiling the Link Between Extreme Precipitation Events and Flood Disasters in China: From a 3D Perspective"

_EGUsphere, 2025_

## Referee Comment (RC1)

Review comments, "Unveiling the Link Between Extreme Precipitation Events and Flood Disasters in China: From a 3D Perspective" by Wang et al.

This paper investigates precipitation space-time characteristics (from IMERG, presumably primarily from GPM and TRMM satellite missions) for 623 flood-causing precipitation (FCP) events over China during the period 2000-2023. Their primary conclusion is that "despite the increase in 3D characteristics of FCP events over the past two decades, flood disasters have shown a significant reduction, except for the direct economic losses". Unfortunately, they never say what they mean by "3D characteristics – the term appears only three times in the paper, twice in the abstract and once in the discussion -- but from Figure 3 one might conclude that these are centroid, magnitude, area, lifespan, moving distance, and moving speed. Importantly, these are all attributes of FCP; none have to do with flood characteristics themselves (e.g., inundation extent, depth, peak discharge, and so on). Their primary source of flood information is something called the China Meteorological Disaster Yearbook. Exactly what is in this publication (or data base) I can't tell for sure as it's In Chinese, but it appears that it's only high-level attributes such as what one might find in NOAA's Billion Dollar Natural Disasters.

So, they've managed to sidestep the entire field of flood hydrology, as well as what arguably is one of the current grand challenges facing the hydrologic community, specifically, if extreme precipitation is increasing, why aren't floods? (they do cite a 2018 WRR paper with that title). There in fact is reasonably conclusive observational evidence that extreme precipitation is increasing, and they cite some key papers along that line. But, they also note that "... despite the observed and projected increase in extreme precipitation, the flood magnitude and frequency exhibit mixed trend patterns. Several global scale flood trend detection studies have found that more stations exhibit significant decreasing trends in flood magnitude than increasing ones ... These inconsistencies highlight the nonlinear relationship between precipitation intensity and flood disasters" What is widely acknowledged is that flood damages (but not necessarily deaths, especially in the developed world) have been increasing. Tanoue et al. (2016) have a pretty good paper on this, showing that increasing flood damages are strongly linked to increased development on flood plains. Surely this must be the case in China, especially during their study period. But that's hardly a new finding.

Given the above, the contribution of this paper isn't clear to me. Certainly, "mining" the meteorological disaster yearbook, along with the precipitation data base, could yield some interesting insights, but I don't see how much that's very interesting could come from that without closely investigating the hydrological aspects of the floods. This clearly would involve some filtering to remove (or account for) the effects of flood regulation by dams and

other means.  There also is a question in my mind as to whether the length of the database (24 years) is sufficient, although perhaps they could make an argument for space for time substitution.  In any event, the absence of any flood hydrology in the current version is, in my view, a fatal flaw, and I think the authors need to go back to the drawing board (and perhaps augment the author group with some flood hydrology expertise).